# DPP-TTS: Diversifying prosodic features of speech via determinantal point processes

## Abstract

With the rapid advancement in deep generative models, recent neural text-to-speech models have succeeded in synthesizing human-like speech, even in an end-to-end manner. However, many synthesized samples often have a monotonous speaking style or simply follow the speaking style of their ground-truth samples. Although there have been many proposed methods to increase the diversity of prosody in speech, increasing prosody variance in speech often hurts the naturalness of speech. Recently, Determinantal point processes (DPPs) have shown remarkable results for modeling diversity in a wide range of machine learning tasks. However, their application in speech synthesis has not been explored. To enhance the expressiveness of speech, we propose DPP-TTS: a text-to-speech model based on a determinantal point process to diversify prosody in speech. The extent of prosody diversity can be easily controlled by adjusting parameters in our model. We demonstrate that DPP-TTS generates more expressive samples than baselines in the side-by-side comparison test while not harming the naturalness of the speech.

## 1 Introduction

In the past few years, text-to-speech models have made a lot of progress in synthesizing human-like speech (Shen et al., 2018; Ping et al., 2018; Li et al., 2019; Ren et al., 2019). Furthermore, in the latest studies, several text-to-speech models made high-quality speech samples even in the end-to-end setting without a two-stage synthesis process (Donahue et al., 2021; Kim et al., 2021). Based on these technical developments, text-to-speech models are now able to generate high-fidelity speech.

However, human speech contains many prosodic features like intonation, stress, and rhythm beyond textual features. Therefore, it is crucial to generate speech samples with rich prosody. There have been many attempts to synthesize speech with rich prosodic features. To incorporate rich prosodic features into speech, reference acoustic samples like mel-spectrograms are processed through reference encoder (Skerry-Ryan et al., 2018; Wang et al., 2018) or text-to-speech models are conditioned on prosodic features like duration, pitch and energy then these features are predicted or manually controlled at inference (Ren et al., 2021; Lańcucki, 2021). However, these methods have drawbacks that the bottleneck dimension should be carefully tuned for desirable results, or the prosody predictor just learns averaged prosodic features of training sets.

Meanwhile, generative models like VAEs and flow models (Hsu et al., 2019; Sun et al., 2020; Vallés-Pérez et al., 2021; Lee et al., 2021; Valle et al., 2021) have been recently used for speech synthesis and their latent spaces are manipulated for generating more expressive speech. Controlling the amount of variation in the speech is achieved by generating samples in Gaussian prior with an adequate temperature, however, it has two major drawbacks. First, generating latent samples with high variance often make generated samples to be unstable in terms of the naturalness and intelligibility of speech. Second, sampling in the latent space might only cover major modes learned during the training stage and only monotonous prosodic patterns can occur in some segments of speech.

Determinantal point processes (DPPs) have shown great results for modeling diversity in various machine learning tasks and their applications. Its uses include text summarization (Cho et al., 2019), recommendation systems (Gartrell et al., 2021), multi-modal output generation (Elfeki et al., 2019), diverse trajectory forecasting (Yuan & Kitani, 2020), and machine translation (Meister et al., 2021). Determinantal point process (DPPs) offers us an efficient subset selection method by considering

both the quality and diversity of items within a set. Specifically, items within a set are sampled according to DPPs kernel which reflects the quality and diversity of items in the ground set. In some cases, we may need to incorporate the conditional information into the sampling process. To this end, conditional DPP (Kulesza & Taskar, 2012) has been widely used for sampling subset by setting the kernel conditioned on the specific information.

To use conditional DPP for sampling prosodic features of speech, two main issues should be considered. First, prosodic features are usually varying in length. Second, it is ambiguous what the ground set is in this problem. In this work, we resolve these issues by using a new similarity metric between prosodic features and adding a prosodic diversifying module (PDM) into the framework. Specifically, PDM generates candidates of prosodic features of speech by mapping latent codes from normal distribution to a new latent space and these candidates are used as the ground set in the sampling process. In addition, the similarity between two prosodic features is evaluated using soft dynamic time warping discrepancy (Soft-DTW) (Cuturi & Blondel, 2017) which enables to evaluate the similarity between features in different lengths.

The kernel matrix of conditional DPP is learned during the training by parameters of PDM getting updated. Specifically, parameters of the prosody diversifying module (PDM) are updated in the training stage with conditional maximum induced cardinality (MIC) objective which is adapted from MIC objective introduced in Gillenwater et al. (2018). We formally introduce the conditional MIC objective and its derivative for clarity in Section 4.

To implement the aforementioned DPP, a stochastic duration predictor and pitch predictor are introduced in this work. Both the duration and pitch predictor are built upon normalizing flows trained with the maximum likelihood (MLE) objective. At inference, the prosody predictor maps latent codes from a latent space to feature spaces with the inverse flow.

In experiments, we compare DPP-TTS with the state-of-the-art models including VITS (Kim et al., 2021) and Flowtron (Valle et al., 2021) in terms of prosody diversity and speech quality. The results demonstrate that our model generates speech with richer prosody than baselines while maintaining speech naturalness. We also demonstrate that our DPP-TTS can be used in real-time applications by evaluating the inference speed of our model.

In summary, our contributions follow:

- We propose a novel method for diversifying prosodic features of speech based on conditional DPPs by considering prosodic features of context words as conditions.
- To learn the kernel matrix of conditional DPPs, we propose to train prosody diversifying module (PDM) with the conditional maximum induced cardinality (MIC) objective.
- Experiments on the side-by-side comparison and the MOS test verify that our model outperforms the two baseline models in terms of prosodic diversity while maintaining the naturalness of speech.

## 2 BACKGROUND

### 2.1 DETERMINANTAL POINT PROCESSES

DPPs encourage diversity within a set by discouraging sampling similar items within the ground set. Formally, point process $\mathcal{P}$ is called a determinantal point process when $\boldsymbol{Y}$ is a random subset drawn according to $\mathcal{P}$, we have, for every $A \subseteq \mathcal{Y}$,

$$\mathcal{P}(A \subseteq \boldsymbol{Y}) \propto \det(\boldsymbol{K}_A),\tag{1}$$

where $\boldsymbol{K}$ is a positive definite matrix whose eigenvalues are all between 0 and 1 and $\boldsymbol{K}_A$ is a positive a definite matrix indexed by elements in $A$. The marginal probability of including two elements $e_i$ and $e_j$ is $K_{ii}K_{jj} - K_{ij}^2 = p(e_i \in \boldsymbol{Y})p(e_j \in \boldsymbol{Y}) - K_{ij}^2$. Therefore, the value of $K_{ij}^2$ models the extent of negative correlation between item $i$ and $j$. More frequently, DPPs are defined by L-ensemble through real and symmetric matrix $\boldsymbol{L}$ instead of the marginal kernel $\boldsymbol{K}$:

$$\mathcal{P}_L(\boldsymbol{Y} = Y) = \frac{\det(\boldsymbol{L}_Y)}{\det(\boldsymbol{L} + \boldsymbol{I})},\tag{2}$$

where $\det(\boldsymbol{L} + \boldsymbol{I})$ in the denominator acts as a normalization constant. The marginal kernel $\boldsymbol{K}$ and kernel $\boldsymbol{L}$ has the following relation:

$$\boldsymbol{K} = \boldsymbol{L}(\boldsymbol{L} + \boldsymbol{I})^{-1} = \boldsymbol{I} - (\boldsymbol{L} + \boldsymbol{I})^{-1} \tag{3}$$

To model diversity between items, the DPP kernel $\boldsymbol{L}$ is usually constructed as a symmetric similarity matrix, where $S_{ij}$ represents the similarity between two items $x_i$ and $x_j$. In Kulesza & Taskar (2010), authors proposed decomposing the kernel $L$ as a a Gram matrix incorporating a quality vector to weigh each item according to its quality :

$$\mathcal{P}(J \subseteq Y) \propto \det(\phi(J)^{\mathrm{T}}\phi(J)) \prod_{e_i \in J} q^2(e_i) = \mathrm{Diag}(\boldsymbol{q}) \cdot \boldsymbol{S} \cdot \mathrm{Diag}(\boldsymbol{q}), \tag{4}$$

where $\phi_i \in \mathbb{R}^D$; $D \leq N$ and $||\phi_i||_2 = 1$. In this manner, the similarity matrix $\boldsymbol{S}$ is guaranteed to be real positive semidefinite matrix.

## 2.2 Conditional determinantal point processes

If DPPs are used for diversifying prosodic features by setting targets of DPPs as prosodic features of sentences, it would result in diversity among generated samples. However, there still can be monotonous patterns in each generated speech. To resolve this issue, it is required to diversify prosodic features accounting into their neighbor prosodic features. Therefore, we need conditional DPPs to take into account neighboring prosodic features (contexts).

By setting conditions of point process $\mathcal{P}$, DPPs can be extended to conditional DPPs, For a subset $B \subseteq Y$ not intersecting with $A$ we have

$$\mathcal{P}(\boldsymbol{Y} = A \cup B | A \subseteq \boldsymbol{Y}) = \frac{\mathcal{P}(\boldsymbol{Y} = A \cup B)}{\mathcal{P}(A \subseteq \boldsymbol{Y})} = \frac{\det(\boldsymbol{L}_{A \cup B})}{\det(\boldsymbol{L} + \boldsymbol{I}_{\bar{A}})}, \tag{5}$$

where $I_{\bar{A}}$ is the matrix with ones in the diagonal entries indexed by elements of $\mathcal{Y} - A$ and zeros elsewhere. In Borodin & Rains (2004), authors showed that this conditional distribution is again a DPP, with a kernel given by

$$\boldsymbol{L}^A = ([(\boldsymbol{L} + \boldsymbol{I}_{\bar{A}})^{-1}]_{\bar{A}})^{-1} - \boldsymbol{I}. \tag{6}$$

In conditional DPPs, items in the ground set $\mathcal{Y}$ are sampled according to kernel given contexts. In this work, contexts of target words are used as conditions for conditional DPPs.

## 2.3 Soft Dynamic time warping

To build the kernel of conditional DPPs, we need a measure of similarity between two temporal signals. Simple Euclidean distance is not applicable because two time signals often vary in their lengths. In this work, soft dynamic time warping (Soft DTW) discrepancy which allows measuring the similarity of shape between time series of different lengths is adapted to calculate the similarity between prosodic sequences. DTW computes the best possible alignment between two temporal signals. First, for the length of $n$ and $m$ signals, DTW computes $n$ by $m$ pairwise distance matrix between these points with a specific metric (e.g., Euclidean distance, L1 distance). After that, this matrix is used to solve the dynamic program using Bellman's recursion with a quadratic $\mathcal{O}(mn)$ cost.

Unfortunately, vanilla DTW cost is not easy to optimize because it only considers a single alignment between two temporal signals. Cuturi & Blondel (2017) proposed differentiable Soft-DTW which considers all possible alignments between two temporal series. Soft-DTW is differentiable in all of its arguments with quadratic cost. Formally, Soft-DTW is defined as follows:

$$\mathbf{dtw}_{\gamma}(\boldsymbol{x}, \boldsymbol{y}) := \min^{\gamma}\{\langle A, \delta(\boldsymbol{x}, \boldsymbol{y})\rangle, A \in \mathcal{A}_{n,m}\}, \tag{7}$$

where $\mathcal{A}_{n,m}$ denotes all possible alignments between $\boldsymbol{x}$ and $\boldsymbol{y}$ and $\min^{\gamma}$ is defined as

$$\min^{\gamma}\{a_1, ..., a_n\} := \begin{cases} \min_{i \leq n} a_i, & \text{if} \quad \gamma = 0 \\ -\gamma \log \sum_{i=1}^{n} e^{\frac{-a_i}{\gamma}}, & \text{otherwise} \end{cases} \tag{8}$$

A small magnitude of $\gamma$ reflects the true discrepancy of two temporal signals, however, optimization becomes unstable.

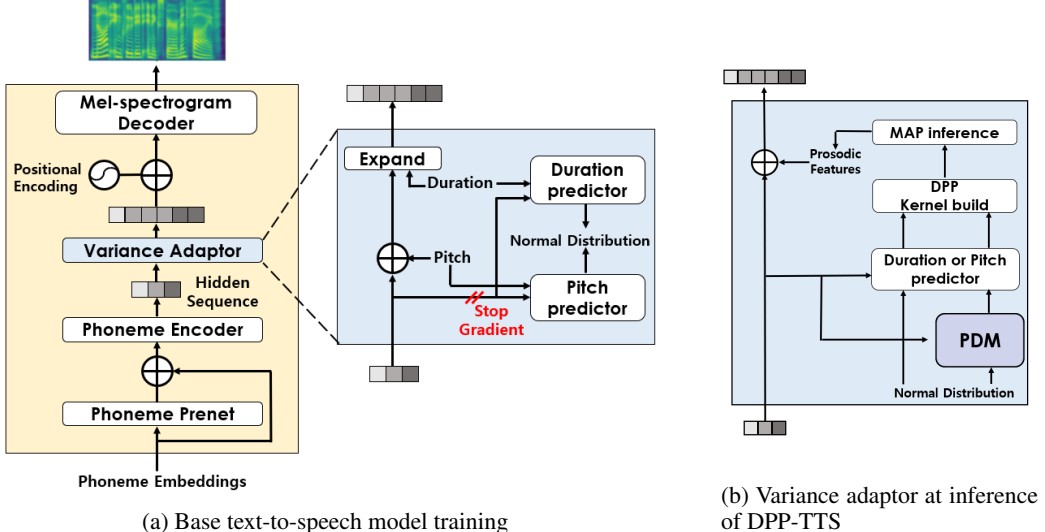

(a) Base text-to-speech model training

(b) Variance adaptor at inference of DPP-TTS

Figure 1: Diagrams describing (a): training of base text-to-speech model and (b) variance adaptor at the inference of DPP-TTS after PDM is added.

## 3 DPP-TTS

Our model DPP-TTS is composed of a Seq2Seq module for generating the mel-spectrogram, a prosody predictor for predicting duration and pitch sequences, and a prosody diversifying module (PDM). At the first stage, the base TTS model which consists of the Seq2Seq module and the prosody predictor is trained as shown in Figure 1a. Once the base TTS is trained, PDM is inserted in front of the prosody predictor and trained with the method which will be described in detail in Section 4. We describe the main modules of DPP-TTS and their roles in the following subsection.

### 3.1 MAIN MODULES OF DPP-TTS

**Seq2Seq module** The role of the Seq2seq module is generating mel-spectrograms from phoneme sequences. The module is adapted from FastSpeech2 (Ren et al., 2021) with some modifications. The model consists of four main parts: a phoneme prenet, a phoneme encoder, a variance adaptor, and a mel-spectrogram decoder. In the phoneme encoder, phoneme sequences are processed through a stack of feed-forward transformer blocks with a relative positional representation (Shaw et al., 2018). In variance adaptor at training, pitch embeddings and energy embeddings are added to encoded hidden representations and then hidden representations are expanded according to ground-truth duration labels [1]. At inference, these prosodic features are provided from predictions of the prosody predictor. Finally, expanded representations are processed through a stack of feed-forward transformer blocks and mel-spectrograms with 80 channels are generated after the linear projection. The Seq2Seq module is trained to minimize $L_1$ distance between the predicted and target mel-spectrogram.

**Prosody predictor** In FastSpeech2, the variance adaptor consists of deterministic predictors for predicting prosodic features. However, a deterministic prosodic predictor is not expressive enough to learn the speaking style of a person. For diverse rhythm and pitch, a stochastic duration predictor and pitch predictor are built upon normalizing flows. Specifically, the stochastic duration predictor estimates the distribution of phoneme duration and the stochastic pitch predictor estimates the distribution of phoneme pitch from the hidden sequence. At the training stage, the prosody predictor learns the mapping from the distribution of prosodic features to normal distribution. At inference, it predicts the phoneme-level duration or pitch by reversing the learned flows. In addition, it also

---

[1]Ground-truth labels are obtained via monotonic alignment search (Kim et al., 2020) between the phonemes and mel-spectrogram.

serves as the density estimator for prosodic features during the training of PDM which will be described in detail in Section 4. The prosody predictor is trained to maximize a variational lower bound of the likelihood of the phoneme duration or pitch. More details regarding the prosody predictor are in Appendix A.

**PDM**    Although the stochastic duration and pitch predictor are trained to generate a speech with diverse rhythm and pitch, the prosody predictor may favors major modes and it leads to the monotonous prosodic pattern in the speech. For more expressive speech modeling, PDM is added in front of the prosody predictor as shown in Figure 1b. Its role is to guide latent codes from a standard normal distribution to another distribution for diverse prosodic features of speech. This module is trained with an objective based on conditional DPPs which is described in Section 2.2. At inference of DPP-TTS, multiple prosodic candidates are generated by PDM and the prosodic feature of speech is selected via MAP inference.

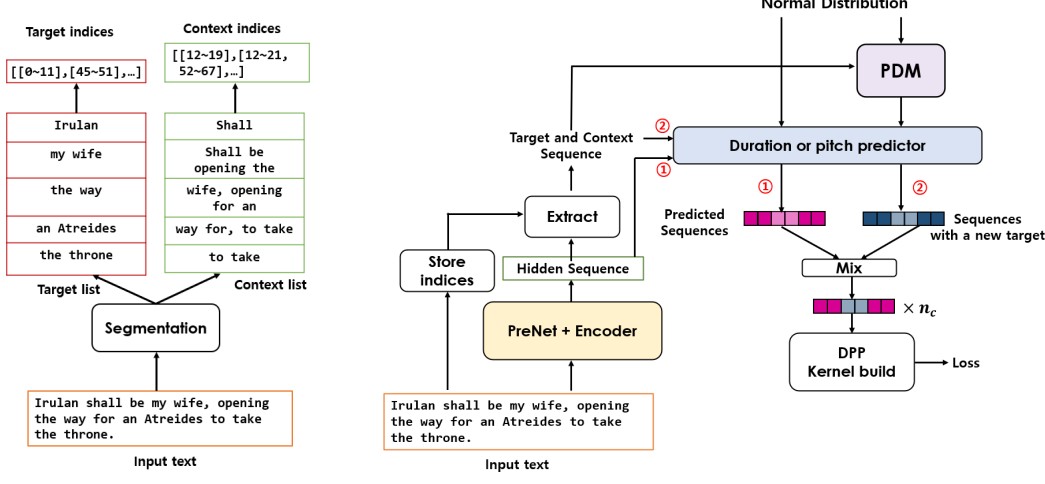

(a) Segmentation of an input text                    (b) Training procedure of PDM

Figure 2: Diagrams describing (a): segmentation of an input text for generating target and context indices and (b) training procedure of PDM.

## 4    DIVERSIFYING PROSODIC FEATURES OF SPEECH WITH PDM

In this section, we explain the methodology for training the prosody diversifying module (PDM). The training process mainly consists of three steps: segmentation of an input text, generation of prosodic feature candidates, building DPP kernel. We also explain the conditional maximum induced cardinality (MIC) objective for training PDM. The overall training procedure is depicted in Figure 2b.

### 4.1    SEGMENTATION OF AN INPUT TEXT

In this stage, targets in input text for diversification of prosody are chosen. The scale of each target is important for the desired goal. Phoneme-level targets would result in too dynamic prosody of generated speech. In contrast, sentence-level targets may result in a monotonous pattern in each speech corresponding to each sentence. Therefore, we choose a few words as each target in the input text. However, naively choosing random words in the input text is not desirable because prosodic boundaries and syntax of the sentence are closely related (Cole et al., 2010). We choose the noun phrases in the input text as each target using Spacy [2] library to take account into the syntactic structure of the input text. Neighboring words with the same number of words as the target are selected as the left of right context. For example, as shown in Figure 2a, if the target is `an Atreides`, then `way for` and `to take` are selected as the left and right contexts respectively. Indices of targets and contexts are stored after the segmentation.

---

[2] https://github.com/explosion/spaCy

## 4.2 GENERATION OF PROSODIC FEATURE CANDIDATES

In this stage, multiple prosodic candidates are generated for the ground set of DPP as shown in Figure 2b. First, an input text is fed into the pre-trained phoneme prenet and encoder and a hidden sequence is generated. Second, the pre-trained prosody predictor conditioned on the hidden sequence predicts the whole prosodic features of the input text from the latent codes samples from a normal distribution. Third, new $n_c$ latent codes from a normal distribution are fed into PDM, then the prosody predictor conditioned on hidden sequence corresponds to the target and its neighboring context predicts new $n_c$ prosodic features. Finally, newly generated target prosodic sequences and previously predicted prosodic sequences are mixed, then $n_c$ prosodic candidates are generated with the target and context entangled.

## 4.3 CONSTRUCTION OF DPP KERNEL

Generated candidates are split into left and right context $d_L, d_R$ and $n$ targets $d_1, d_2, ...d_n$, then the ground set for DPP is constructed merging the targets and contexts as shown in Figure 3. The kernel of conditional DPP is built incorporating both diversity and quality of the ground set. As mentioned in Section 2.1, the kernel of DPP can be decomposed as $L = \text{Diag}(\boldsymbol{q}) \cdot S \cdot \text{Diag}(\boldsymbol{q})$, where $q$ denotes the quality vectors of predicted features and $S$ denotes the similarity matrix.

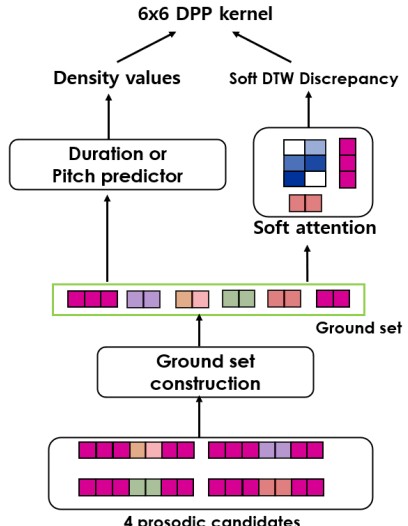

Figure 3: Construction of DPP kernel with $n_c = 4$

**Similarity metric**  Target sequences and context sequences often vary in lengths, therefore the Euclidean distance is not applicable to calculate the similarity between two sequences. The similarity metric is calculated as follows based on Soft DTW:

$$\boldsymbol{S}_{i,j} = \exp(-k \cdot \textbf{dtw}_\gamma^D(\boldsymbol{d}_i, \boldsymbol{d}_j)) \qquad (9)$$

$\textbf{dtw}_\gamma^D$ denotes soft-DTW discrepancy with a metric $D$ and smoothing parameter $\gamma$. When the metric $D$ is chosen as the $L_1$ distance $D(\boldsymbol{x}, \boldsymbol{y}) = \sum_i |\boldsymbol{x}_i - \boldsymbol{y}_i|$ or half gaussian $D(\boldsymbol{x}, \boldsymbol{y}) = ||\boldsymbol{x} - \boldsymbol{y}||_2^2 + \log(2 - \exp(-||\boldsymbol{x} - \boldsymbol{y}||_2^2))$, the similarity matrix is positive semi-definite (Cuturi et al., 2007; Blondel et al., 2021). In this work, $L_1$ distance is used as the metric of Soft-DTW so that $\boldsymbol{S}$ to be positive semi-definite.

**Quality metric**  Based on the estimated density of predicted features, quality scores are calculated. The density values of predicted features are calculated with importance sampling using the prosody predictor: $p(\boldsymbol{x}; \theta) \approx \sum_{i=1}^{N} \frac{p(\boldsymbol{x}, \boldsymbol{z}_i; \theta)}{q(\boldsymbol{z}_i | \boldsymbol{x}; \phi)}$. In the experiment, using raw density value resulted in not enough diverse sample, therefore quality values are not penalized if the density value is greater than a specific threshold. With log likelihood $\pi(\boldsymbol{x}) = \log p(\boldsymbol{x})$, the quality score is defined as

$$\boldsymbol{q}(\boldsymbol{x}) = \begin{cases} w & \text{if } \pi(\boldsymbol{x}) >= k \\ w \cdot \exp(\pi(\boldsymbol{x}) - k) & \text{otherwise} \end{cases} \qquad (10)$$

The threshold value $k$ was set as the average density of the dataset in the experiment. Finally, the kernel of conditional DPPs is defines as $\boldsymbol{L} = \text{diag}(q) \cdot \boldsymbol{S} \cdot \text{diag}(q)$.

## 4.4 OBJECTIVE FUNCTION

We need a measure of diversity with respect to kernel $\boldsymbol{L}$ to train the PDM. One straightforward choice is maximum likelihood (MLE) objective, $\log \mathcal{P}_{\boldsymbol{L}}(\boldsymbol{Y}) = \log \det(\boldsymbol{L}) - \log \det(\boldsymbol{L_Y} + I)$. However, there are some cases where almost identical prosodic features are predicted. This leads the objective value to be close to zero which makes the training process unstable. Instead, maximum induced cardinality (MIC) (Gillenwater et al., 2018) objective which is defined as $\mathbb{E}_{\boldsymbol{Y} \sim \mathcal{P}_{\boldsymbol{L}}}[|\boldsymbol{Y}|]$ can be an alternative. It does not suffer from training instability since identical items in the ground

set $\mathcal{Y}$ are allowed. In this work, conditional MIC objective is used as setting contexts $d_L, d_R$ as the condition. The objective function with respect to the ground set $[d_L, d_R, d_1, d_2, ..., d_N]$ and its derivative are as follows:

**Proposition 1** (**MIC objective of conditional DPPs**). *With respect to the ground set $[d_L, d_R, d_1, d_2, ..., d_N]$, the MIC objective of conditional DPPs and its derivative are as follows:*

$$\boldsymbol{L}_{MIC} = \mathrm{tr}(\boldsymbol{I} - [(\boldsymbol{L}_\theta + \boldsymbol{I}_{\bar{A}})^{-1}]_{\bar{A}}), \frac{\partial \boldsymbol{L}_{MIC}}{\partial \theta} = ((\boldsymbol{L} + \boldsymbol{I}_{\bar{A}})^{-1} \boldsymbol{I}_{\bar{A}} (\boldsymbol{L} + \boldsymbol{I}_{\bar{A}})^{-1})^{\mathrm{T}} \boldsymbol{L}'(\theta), \quad (11)$$

*where $A$ denotes the set of contexts $(d_L, d_R)$, $\bar{A}$ denotes the complement of the set $A$ and $\theta$ is the parameter of PDM.*

Detailed proof is presented in Appendix B. At inference, for predicting a single prosodic feature, MAP inference is performed across sets with just a single item as follows: $\boldsymbol{x}^* = \arg\max_{\boldsymbol{x} \in \bar{A}} \log \det(\boldsymbol{L}_{\{\boldsymbol{x}\} \cup A})$. The detailed procedure of training PDM and the inference of DPP-TTS are in Appendix C.

## 5 EXPERIMENTS

In this section, we first explain the experimental setup and then human evaluation results and further evaluation results that are conducted to evaluate the proposed DPP-TTS. Audio samples used for the evaluation are in the supplementary material.

### 5.1 EXPERIMENTAL SETUP

**Dataset and preprocessing**  We conducted experiments on the LJ Speech dataset which consists of audio clips with approximately 24 hours lengths. We split audio samples into 12500/100/500 samples for the training, validation, and test set. Audio samples with 22kHz sampling rate are transformed into 80 bands mel-spectrograms through the Short-time Fourier transform (STFT) with 1024 window size and 256 hop length. International Phonetic Alphabet (IPA) sequences are used as input for phoneme encoder. Text sequences are converted to IPA phoneme sequences using `Phonemizer`[3] software. Following Kim et al. (2020), the converted sequences are interspersed with blank tokens, which represent the transition from one phoneme to another phoneme.

**Training Setup**  Both the basic TTS and PDM are trained using the AdamW optimizer (Loshchilov & Hutter, 2019) with $\beta_1 = 0.8, \beta_2 = 0.99$ and $\lambda = 0.01$. The initial learning rate is $2 \times 10^{-4}$ for the basic TTS and $1 \times 10^{-5}$ for the PDM and the exponential learning decay is used for both of them. Basic TTS is trained with 16 batch size for 200k steps on 2 NVIDIA RTX 3080 GPUs. PDM is trained with 8 batch size for 5k steps on same devices. In both the duration and pitch diversification modeling, the quality weight of the model is set as $w = 10$.

**Baseline**  We compare our model with two state-of-the-art models in terms of naturalness and prosody diversity. The first one is VITS (Kim et al., 2021) which is an end-to-end TTS model based on conditional VAE and normalizing flows. In all experiments, the temperature $\sigma$ of the duration predictor in VITS was set 0.8. The second one is Flowtron (Valle et al., 2021) which is an autoregressive flow-based TTS model. In experiments, the temperature $\sigma$ of Flowtron was set to 0.6. For our DPP-TTS and Flowtron, HiFi-GAN (Kong et al., 2020) is used as the vocoder for synthesizing waveforms from the mel-spectrograms.

### 5.2 HUMAN EVALUATION RESULTS

**Side by side evaluation**  First, we conduct a side-by-side evaluation of the prosody of a sentence. In this experiment, texts from test samples from LJSpeech are synthesized. Target phrases extracted from the sentence are processed by PDM and pitch diversity modeling is used in DPP-TTS. Via Amazon Mechanical Turk (AMT), we assign five testers living in the United States to a pair of audio samples (i.e., DPP-TTS and a baseline), and ask them to listen to audio samples and choose among three options: **A**: sample A has more varied prosody than sample B, **Same**: sample A and B are equally varied in prosody, **B**: the opposite of option (1).

---

[3]https://github.com/bootphon/phonemizer

In addition, we also conduct a side-by-side evaluation of prosody in a paragraph that is not included in the LJSpeech dataset since a monotonous pattern is prominent in long texts rather than a single sentence. We prepare two DPP-TTS versions: (1): DPP-TTS1: a model that has only duration diversifying module and DPP-TTS2: a model that has both duration and pitch diversifying modules. In this experiment, only VITS is compared as the baseline. Likewise the evaluation of a sentence, listeners are asked to choose among the three options.

Results are shown in Table 1, 2. In the side-by-side evaluation of prosody in a sentence, DPP-TTS outperforms VITS and Flowtron. In the side-by-side evaluation of prosody in a paragraph, both the DPP-TTS1 and DPP-TTS2 outperform the baseline.

Table 1: Side-by-side comparison on the LJSpeech dataset.

| Model A/B | A | Same | B |
|---|---|---|---|
| DPP-TTS/Flowtron | 55.3% | 12.7% | 32.0% |
| DPP-TTS/VITS | 40.0% | 32.0% | 28.0% |

Table 2: Side-by-Side comparison on the paragraph.

| Model A/B | A | Same | B |
|---|---|---|---|
| DPP-TTS1/VITS | 56.25% | 6.25% | 37.5% |
| DPP-TTS2/VITS | 50.0% | 12.5% | 37.5% |

**Naturalness test** In this experiment, we measure the Mean-Opinion-Score (MOS) for audio samples evaluated in the side-by-side evaluation. Likewise in the previous test, five testers are assigned to each audio sample. Testers are asked to give a score between 1 to 5 on a 9-scale based on the sample's naturalness. We used the same quality weight value $w = 10$ for DPP-TTS as in the side-by-side comparison test. **MOS-S** denotes the quality of audio from the LJSpeech dataset and **MOS-P** denotes the quality of audio from the paragraph. The MOS result is shown in Table 3. In MOS-S, our model gets lower MOS than VITS, it yet still maintain a MOS of 3.92 and outperforms Flowtron. In MOS-P, our model outperforms VITS. A significant reduction of MOS in MOS-P compared to MOS-S indicates that it's challenging to maintain the naturalness in long texts.

Table 3: Mos evaluation results with 95% confidence intervals.

| Model | MOS-S | MOS-P |
|---|---|---|
| VITS | $4.08 \pm 0.53$ | $3.06 \pm 0.52$ |
| Flowtron | $3.81 \pm 0.72$ | - |
| DPP-TTS | $3.92 \pm 0.58$ | $3.13 \pm 0.54$ |
| Ground-truth | $4.46 \pm 0.51$ | - |

## 5.3 ADDITIONAL EVALUATION

In this section, we conduct additional evaluations for our DPP-TTS and VITS [4]. We use the following metrics for the evaluation : (1) $\sigma_{frame}$: frame-level standard deviation of the pitch in a speech. All pitch values are normalized since speech samples have different absolute pitch values. Standard deviations are averaged over 50 generated samples. DIO algorithm (Morise et al., 2009) is used for pitch extraction. (2): $\sigma_{phoneme}$: phoneme-level standard deviation of duration or pitch in a speech. Likewise, the standard deviation values are averaged over 50 generated samples. (3): **Determinant**: a determinant of the matrix with respect to a feature set $J$ generated by multiple samples is used to evaluate the diversity of phoneme-level prosodic features of multiple generated samples. Cosine similarity is used as a metric between two sequences. Each set is constructed from 10 generated samples. (4): **Inference time**: the inference time for synthesizing one-second length waveform is calculated in the text-to-speech model. The inference speed is evaluated on Intel(R) Core(TM) i7-7800X CPU and a single NVIDIA RTX 3080 GPU. Computation time is averaged over 100 forward passes.

| Model | Pitch $\sigma_{frame}$ | Pitch $\sigma_{phoneme}$ | Pitch Determinant | Duration $\sigma_{phoneme}$ | Duration Determinant | Inference Time(s) |
|---|---|---|---|---|---|---|
| DPP-TTS | **0.012** | **0.014** | **$2.92 \times 10^{-14}$** | **1.81** | **$7.21 \times 10^{-7}$** | $4.4 \times 10^{-2}$ |
| VITS | 0.010 | 0.011 | $5.58 \times 10^{-17}$ | 0.22 | $6.68 \times 10^{-7}$ | **$1.3 \times 10^{-2}$** |

Table 4: Additional evaluation results. Numbers in bold denote the better result.

Table 4 shows the result of $\sigma_{frame}$, $\sigma_{phoneme}$, determinant and inference time. At both frame and phoneme levels, the standard deviation of the pitch in DPP-TTS results in higher values than the baseline. In addition, the standard deviation of duration at phoneme-level in DPP-TTS also results in a higher value than the baseline. It demonstrates that DPP-TTS generates a speech with more

---

[4]Hyperparameter settings remain same as the Section 5.2.

dynamic pitch and rhythm than the baseline. The determinants of duration and pitch sets of DPP-TTS outperform the baseline. It shows that DPP-TTS generates more diverse samples in terms of prosody than baseline. Finally, DPP-TTS results in 0.044 seconds of inference speed. Although its inference speed is slower than the baseline, our model is applicable in practice since the inference speed of our model is 22.7x faster than real-time.

## 5.4 MODEL ANALYSIS

**Ablation study** We conduct ablation studies to verify the effectiveness of PDM, the results are shown in Table 5. In the side-by-side comparison, 34% more listeners choose the option that DPP-TTS has more varied prosody compared to DPP-TTS without PDM. It demonstrates that PDM contributes to prosody diversification significantly. DPP-TTS also achieves a higher phoneme-level standard deviation of pitch and duration than the baseline. Although there is a quality degradation (-0.26 MOS) compared to the baseline, DPP-TTS still achieve a MOS of 3.92.

| Model | Side-by-Side Comparison | MOS | Pitch $\sigma_{phoneme}$ | Duration $\sigma_{phoneme}$ |
|---|---|---|---|---|
| DPP-TTS | 58.5% | 3.92±0.58 | 0.014 | 1.81 |
| DPP-TTS w/o PDM | 17.0% 24.5% | 4.18 ± 0.49 | 0.01 | 1.76 |

Table 5: Comparison in the ablation studies.

**Adjusting the extent of variation** To study the impact of quality weight $w$, we plot (normalized) pitch and log duration values at phoneme-level with different values of quality weight. As expected, for a large magnitude of quality weight , the model learns diversifying prosodic features with small deviations from contexts. In contrast, for a small magnitude of quality weight , the model diversifies prosodic features with large deviations from contexts. We also found that a small magnitude of quality weight leads to faster convergence of the model but it easily hurts the naturalness of generated samples. Figures are shows in Appendix E.

**Effect of number of candidates on diversity** In this experiment, we study the effect of the number of candidates on the prosody diversity in generated samples. For a different number of candidates $n_c = 4, 8, 12$, DPP-TTS is trained up to 4k steps and all other hyperparameters remain unchanged. After training, we calculate pitch

| Number of candidates | Pitch determinant |
|---|---|
| $n_c = 4$ | $1.69 \times 10^{-15}$ |
| $n_c = 8$ | $4.56 \times 10^{-15}$ |
| $n_c = 12$ | $\mathbf{7.8 \times 10^{-15}}$ |

Table 6: Pitch determinant values with different number of candidates

determinants over generated samples for each setup. The result is reported in Table 6. We observe that using a large number of candidates results in more diverse samples although only a single prosody item is selected at MAP inference. We also find that using a large number of candidates has a regularization effect for training. Using a small number of candidates generates unstable speech samples in terms of naturalness, in contrast, using a large number of candidates results in more stable speech samples.

## 6 CONCLUSION

In this paper, we have proposed a novel prosody diversifying method based on conditional determinantal point processes (DPPs). To learn the kernel matrix of conditional DPPs, conditional determinantal point process (DPPs) is parameterized additional prosody diversifying module (PDM). To build the kernel, Soft dynamic time warping is adopted to measure the extent of similarity between two prosodic features, and the kernel of conditional DPPs is learned with conditional maximum induced cardinality (MIC) objective. In the experiment, we demonstrated that our new prosody diversifying method generates more dynamic samples than the baseline while maintaining the naturalness. In addition, we demonstrate that our model can be used in real-time applications by showing the inference speed of our model.

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

# A    DETAILS OF PROSODY PREDICTOR AND PDM

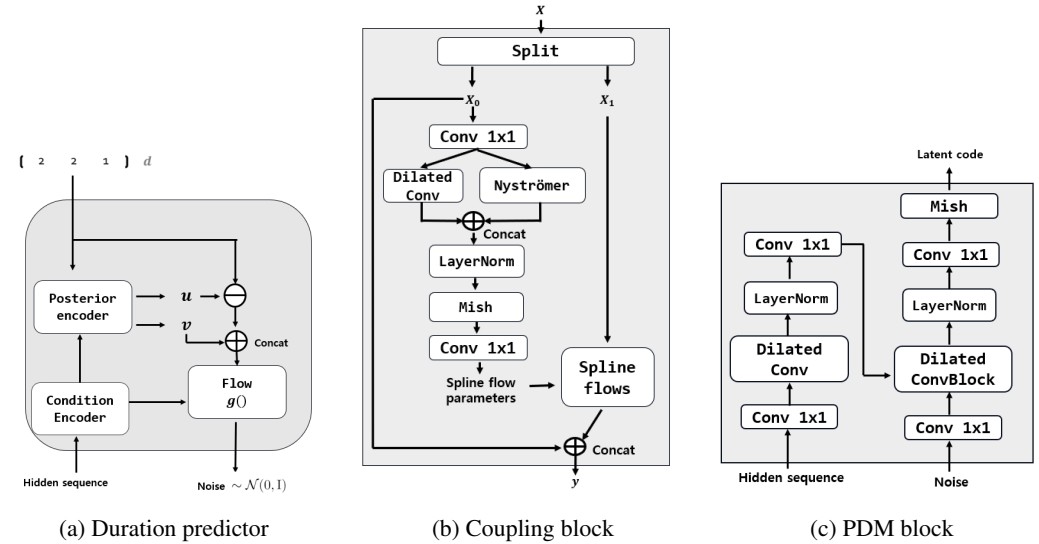

(a) Duration predictor          (b) Coupling block          (c) PDM block

Figure 4: Diagrams describing (a): the duration predictor, (b): Coupling block in normalizng flows and (c): Prosody diversifying module

## A.1    PROSODY PREDICTOR

**Training**    It is hard to use the maximum likelihood objective directly to train duration predictor because the duration of each phoneme is 1) a discrete integer and a scalar, which hinders expressive transformation because invertibility should remain in normalizing flow. To train duration predictor, duration values are extended to continuous values using variational dequantization (Ho et al., 2019) and are augmented with extra dimensions using variational data augmentation (Chen et al., 2020). Specifically, duration sequence $d$ becomes continuous as $d - u$ where $u$'s value is restricted to $[0, 1)$ and augmented as $[d - u, v]$ with a extra random variable $v$. Two random variables $u$ and $v$ are sampled through approximate posterior $q_\phi(u, v | d, h_{text})$. The ELBO can be calculated as follows:

$$\log p_\theta(d|h_{text}) \geq \mathbb{E}_{q_\phi(u,v|d,h_{text})}[\log \frac{p_\theta(d-u, v|h_{text})}{q_\phi(u, v|d, h_{text})}] \tag{12}$$

Like the duration predictor, the ELBO for pitch predictor can be calculated as follows:

$$\log p_\theta(p|h_{text}) \geq \mathbb{E}_{q_\phi(v|p,h_{text})}[\log \frac{p_\theta(p, v|h_{text})}{q_\phi(v|p, h_{text})}], \tag{13}$$

where $p$ denotes the pitch sequences. Both the duration and pitch predictor are trained on negative lower bound of likelihood.

**Architecture**    The pitch predictor and duration predictor share identical architecture except for the additional random variable $u$. We will introduce the architecture of the duration predictor whose diagram is shown in Figure 4a. Duration predictor consists of condition encoder for hidden sequence, posterior encoder for the duration sequence, and the main flow blocks $g(\cdot)$. Specifically, posterior encoder maps latent codes from normal distribution to random variable $[u, v]$ and flow $g(\cdot)$ maps $[d - u, v]$ to normal distribution. Figure 4b shows the coupling of normalizing flows. First, input $x$ is split into $[x_0, x_1]$ and then $x_0$ is processed by a 1x1 convolution block. The output of the convolution block is processed by dilated convolution and Nystromer block and their outputs are concatenated. The concatenated output is processed by LayerNorm, Mish activation, and Convolution block. Finally, spline flows (Durkan et al., 2019) are parameterized by the output and then the output of spline flows $y$ and $x_0$ are concatenated.

## A.2    PDM

The architecture of PDM is shown in Figure 4c. First, the hidden sequence is processed by convolution block followed by the dilated convolution block conditioned on hidden sequences, Layernorm,

and 1x1 convolution block. After that, noise from a normal distribution is processed by 1x1 convolution followed by dilated convolution block, LayerNorm, and Mish activation.

As the prosody pattern of human speech is correlated with the local context and global context of the text, we design the architecture of the model to encode both the local and global features. Specifically, dilated convolution blocks are stacked to encode local features and a transformer block is used to encode global features in the coupling layers of normalizing flows. However, using the vanilla transformer to encode the global features in the coupling layer requires too large computational complexity. Therefore, Nyströmformer (Xiong et al., 2021) which is a Nyström-Based algorithm for approximating self-attention is used to encode global context for more efficient memory usage in the coupling layer of normalizing flow. Encoded local features and global features are concatenated and processed through LayerNorm (Ba et al., 2016), Mish activation (Misra, 2019) and a 1x1 convolution module.

Following the duration predictor model in Kim et al. (2021), variational dequantization (Ho et al., 2019) is used for the duration predictor since the phoneme duration is a discrete integer. In addition, variational augmentation (Chen et al., 2020) is used to expand channel dimensions for expressive flows in both the duration and pitch predictor. The stochastic and pitch predictor are trained by maximizing their variational lower bounds (ELBO). Details of the duration and pitch predictor are described in Appendix A.

## B PROOF OF **Proposition 1**

**Objective**   From equation [45] in (Kulesza & Taskar, 2012), the marginal kernel of conditional DPPs given the appearance of set $A$ has a following form:

$$\boldsymbol{K}^A = \boldsymbol{I} - [(\boldsymbol{L} + \boldsymbol{I}_{\bar{A}})^{-1}]_{\bar{A}} \tag{14}$$

In addition, from equation [34] in Kulesza & Taskar (2012) the expected cardinality of $\boldsymbol{Y}$ given the marginal kernel $\boldsymbol{K}$ is:

$$\mathbb{E}[|\boldsymbol{Y}|] = \sum_{n=1}^{N} \frac{\lambda_n}{\lambda_n + 1} = \mathrm{tr}(\boldsymbol{K}) \tag{15}$$

From equation 14, 15, the expected cardinality of conditional DPPs given the appearance of set $A$ is:

$$\mathbb{E}[|\boldsymbol{Y}|] = \mathrm{tr}(\boldsymbol{K}^{\mathrm{A}}) = \mathrm{tr}(\boldsymbol{I} - [(\boldsymbol{L} + \boldsymbol{I}_{\bar{A}})^{-1}]_{\bar{A}}) \tag{16}$$

**Derivative**   For the proof, we will start with a following lemma:

**Lemma 1.** *Given a matrix $E$ and non-singular matrix $A$, following equation holds:*

$$\frac{\partial}{\partial A} \mathrm{tr}(E^{\mathrm{T}} A^{-1} E) = -(A^{-1} E E^{\mathrm{T}} A^{-1})^{\mathrm{T}} \tag{17}$$

*Proof.* First, consider $\frac{\partial}{\partial A_{ij}} \mathrm{tr}(E^{\mathrm{T}} A^{-1} E)$. Since *trace* and derivative operator are interchangeable,

$$\begin{aligned} \frac{\partial}{\partial A_{ij}} \mathrm{tr}(E^{\mathrm{T}} A^{-1} E) &= \mathrm{tr}(\frac{\partial}{\partial A_{ij}} (E^{\mathrm{T}} A^{-1} E)) \\ &= -\mathrm{tr}(E^{\mathrm{T}} A^{-1} \frac{\partial A}{\partial A_{ij}} A^{-1} E) \end{aligned} \tag{18}$$

By setting $\frac{\partial A}{\partial A_{ij}} = E^{ij}$ where $E^{ij}$ denotes the matrix whose $(i, j)$ component is 1 and 0 elsewhere and $\boldsymbol{C} = -E^{\mathrm{T}} A^{-1} E^{ij} A^{-1} E$,

$$
\begin{aligned}
-\operatorname{tr}(E^{\mathrm{T}} A^{-1} E^{ij} A^{-1} E) &= \sum_{i'} C_{i'i'} \\
&= -\sum_{i'} \sum_{k_1} \sum_{k_2} (E^{\mathrm{T}} A^{-1})_{i'k_1} E^{ij}_{k_1 k_2} (A^{-1} E)_{k_2 i'} \\
&= -\sum_{i'} (E^{\mathrm{T}} A^{-1})_{i'i} (A^{-1} E)_{ji'} = \\
&= -\sum_{i'} (A^{-\mathrm{T}} E)_{ii'} (E^{\mathrm{T}} A^{-\mathrm{T}})_{i'j} = -(A^{-\mathrm{T}} E E^{\mathrm{T}} A^{-\mathrm{T}})_{ij} \\
&= -(A^{-1} E E^{\mathrm{T}} A^{-1})^{\mathrm{T}}_{ij}
\end{aligned} \tag{19}
$$

$$
\implies \frac{\partial}{\partial A_{ij}} \operatorname{tr}(E^{\mathrm{T}} A^{-1} E) = -(A^{-1} E E^{\mathrm{T}} A^{-1})^{\mathrm{T}}_{ij}.
$$

Now, with respect to set a $A$ whose cardinality is $p$ and matrix $\boldsymbol{L} \in \mathbb{R}^{(p+q) \times (p+q)}$

$$
\begin{aligned}
\frac{\partial}{\partial \theta} &\operatorname{tr}(\boldsymbol{I} - [(\boldsymbol{L} + \boldsymbol{I}_{\bar{A}})^{-1}]_{\bar{A}}) \\
&= -\frac{\partial}{\partial \theta} \operatorname{tr}([(\boldsymbol{L} + \boldsymbol{I}_{\bar{A}})^{-1}]_{\bar{A}}) \\
&= -\frac{\partial}{\partial \theta} \operatorname{tr}(E^{\mathrm{T}} (\boldsymbol{L} + \boldsymbol{I}_{\bar{A}})^{-1} E),
\end{aligned} \tag{20}
$$

where $E$ denotes $\begin{bmatrix} \mathbf{0} \\ \boldsymbol{I}_q \end{bmatrix} \in \mathbb{R}^{(p+q) \times q}$. Then by Lemma 1.

$$
\begin{aligned}
-\frac{\partial}{\partial \theta} \operatorname{tr}(E^{\mathrm{T}} (\boldsymbol{L} + \boldsymbol{I}_{\bar{A}})^{-1} E) &= ((\boldsymbol{L} + I_{\bar{A}})^{-1} E E^{\mathrm{T}} (\boldsymbol{L} + \boldsymbol{I}_{\bar{A}})^{-1})^{\mathrm{T}} \boldsymbol{L}'(\theta) \\
&= ((\boldsymbol{L} + I_{\bar{A}})^{-1} \boldsymbol{I}_q (\boldsymbol{L} + \boldsymbol{I}_{\bar{A}})^{-1})^{\mathrm{T}} \boldsymbol{L}'(\theta)
\end{aligned} \tag{21}
$$

The proof of Proposition 1 is now finished.

# C  ALGORITHMS FOR PDM TRAINING AND INFERENCE OF DPP-TTS

## C.1  PDM TRAINING

---
**Algorithm 1** Training of PDM

---
**Require:** TextEncoder $f(\cdot)$, PDM parameterized $\theta$, a prosody predictor (trained flow) $g(\cdot)$, number of candidates $n_c$, noise scale $\epsilon$

1: **while** not converged **do**
2:    $h_{text} \leftarrow f(text)$
3:    Split $h_{text}$ into $[h_{target}, h_{context}]$
4:    Sample latent code $z_{context} \in \mathbb{R}^T$ with noise scale $\epsilon$
5:    Get prosodic features of contexts: $d_{context} \leftarrow g^{-1}(h_{context}, z_{context})$
6:    Get quality of contexts: $q_{context} = $ `Density estimation`$(d_{context})$
7:    Sample latent codes $z_{target}$ with noise scale $\epsilon$
8:    Get latent codes after PDM: $z_{target} \leftarrow \text{PDM}(z_{target}) \in \mathbb{R}^{n_c \times T}$
9:    Get $n_c$ prosodic features of targets: $(d^1_{target}, d^2_{target}, ... d^{n_c}_{target})$
10:   Get quality of targets: $q_{target} = $ `Density estimation`$(d_{target})$
11:   Concatenate contexts and targets: $[q_{context}, q_{target}], [d_{context}, d_{target}]$
12:   Build the kernel of conditional DPPs:
       $\boldsymbol{L} \leftarrow $ Build kernel$([q_{context}, q_{target}], [d_{Context}, d_{target}])$
13:   Calculate the loss function: $L_{diversity} \leftarrow -\text{tr}(I - [(L + I_{\bar{A}})^{-1}]_{\bar{A}})$
14:   Update $\theta$ with the gradient $\nabla L_{diversity}$
15: **end while**

---

## C.2  INFERENCE OF DPP-TTS

---
**Algorithm 2** Inference of DPP-TTS

---
**Require:** TextEncoder $f(\cdot)$, Decoder $h(\cdot)$, PDM, a prosody predictor $g(\cdot)$, noise scale $\epsilon$

1: $h_{text} \leftarrow f(text)$
2: Split $h_{text}$ into $[h_{target}, h_{context}]$
3: Sample latent code $z_{context} \in \mathbb{R}^{rmT}$ with noise scale $\epsilon$
4: Get prosodic features of contexts: $d_{context} \leftarrow g^{-1}(h_{context}, z_{context})$
5: Get quality of contexts: $q_{context} = $ `Density estimation`$(d_{context})$
6: Sample latent codes $z_{target}$ with noise scale $\epsilon$
7: Get latent codes after PDM: $z_{target} \leftarrow \text{PDM}(z_{target}) \in \mathbb{R}^{n_c \times T}$
8: Get $n_c$ prosodic features of targets: $(d^1_{target}, d^2_{target}, ... d^{n_c}_{target})$
9: Get quality of targets: $q_{target} = $ `Density estimation`$(d_{target})$
10: Concatenate contexts and targets: $[q_{context}, q_{target}], [d_{context}, d_{target}]$
11: Build the kernel of conditional DPPs:
     $\boldsymbol{L} \leftarrow $ Build kernel$([q_{context}, q_{target}], [d_{Context}, d_{target}])$
12: Perform the MAP inference $d^* \leftarrow \arg\max_d \text{ logdet}(\boldsymbol{L}_{d \cup d_{context}})$
13: Synthesize wavs with prosodic features: $y \leftarrow h(h_{text}, d^*)$

---

## D  SAMPLE PARAGRAPH FOR THE SIDE-BY-SIDE COMPARISON TEST

```
 Known individually and collectively as Shai-Hulud, the sandworms are
these supermassive beings that plow through the deserts of Arrakis,
consuming everything that dares venture unprepared into their terri-
tory.  The worms are what make harvesting spice so difficult because
they tend to eat whatever tools off-worlders use to mine it. They are
also sacred to the Fremen, who seem to know ways to navigate around
them, and, somehow, they're linked to the creation of spice. Think of
them as big honking metaphors for the sublime powers of nature that
loom beyond human understanding, like a desert full of Moby Dicks
```

## E  ADJUSTING THE EXTENT OF VARIATION

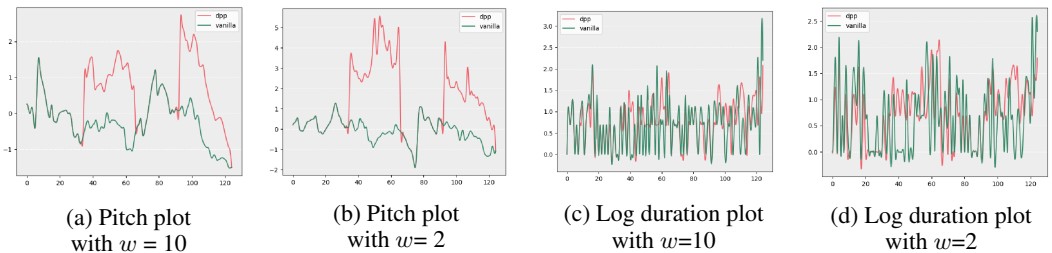

(a) Pitch plot
with $w = 10$

(b) Pitch plot
with $w = 2$

(c) Log duration plot
with $w=10$

(d) Log duration plot
with $w=2$

Figure 5: Pitch and log duration plots with different values of quality weight. Green plots indicate the prosody prediction before MAP inference of DPP and red plots indicate the prosody prediction after MAP inference of DPP.

