# OpenReview forum: "DPP-TTS: Diversifying prosodic features of speech via determinantal point processes"
_ICLR.cc/2022/Conference — ICLR 2022 Submitted_

### Official Review · Reviewer_Dfaw · 2021-11-01

**Correctness:** 2
**Technical Novelty And Significance:** 2
**Empirical Novelty And Significance:** 3
**Recommendation:** 5
**Confidence:** 4

**Main Review:**

Strengths
1. The idea of formulating prosody generation as a determinantal point process (DPP) is novel to the best of my knowledge. DPP indeed has nice properties that explicitly encourage diversity when generating a subset of samples.
2. The design of the similarity metric (soft-DTW) is reasonable and addresses the issue of comparing sequences with varying lengths

Weaknesses
1. Writing requires substantial improvement. Many symbols are undefined when they are first mentioned, notations are messy, and there are a few potential errors in the equations, making it even harder to comprehend. Descriptions of the proposed methods are unclear in several places, and details of how prosody selection fits into the conditional DDP formulation is missing. Please see the detailed comments and questions below for writing issues.
2. DPP is useful for selecting a high quality and diverse subset. Hence, I was expecting the paper to show how this approach can outperform VAE of flow-based methods that draw i.i.d. samples from the latent space both in quality and in diversity when **generating multiple samples** conditioned on the same text. Unfortunately, this capability was not demonstrated empirically: as Section 4.3 shows, only one prosody feature is selected during inference.
3. There is a more serious concern about how conditional DDP is deployed for subset selection. Based on the description of Section 4.3, the goal of applying DDP is to select a subset that includes d_L, d_R and one candidate from d_i where i \in [1, N], such that the subset is high quality and diverse. In other words, given d_L and d_R, the authors want to find a d_i that is of high quality and different from the context, which does not make much sense from the speech synthesis point of view. The context should be coherent with the target, not different. A more reasonable way to apply DDP would be selecting K samples from a set of N candidates which are the prosodic features for the same piece of text. The original design makes s-DTW an unreasonable choice, because DTW aims to align temporal signals corresponding to the same content but may be non-uniformly distorted temporally.
4. The authors argue that previous work such as GST-Tacotron and VAE-based methods require careful parameter tuning (bottleneck or variance). However, the proposed DDP-TTS also requires tuning the weight of the quality term $w$, which does not seem to solve the issues mentioned in the previous work.
5. The authors state “The results demonstrate that our model generates speech with richer prosody than baselines while maintaining speech naturalness...” in the introduction. This is clearly an overstatement in terms of speech naturalness - the proposed model is 0.84 behind VITS on MOS score, which apparently are not of the same level of naturalness.
6. Ablation studies are missing. The authors should have compared with a baseline of their system that does not use the DDP module. More specifically, the authors could sample from the latent distribution (normal) or take the mode of the prior to generate prosodic features.

Questions/Comments
1. The concept of “ground set” is never explicitly defined. The author could have mentioned that as a set of candidate prosodies.
2. For every target sequence, does the prosody predictor encode the left and right context separately?
3. Why conditional DPP is required is not clear when reading Section 2.2. The authors should draw more connections between (conditional) prosody selection to (conditional) DDP when writing Section 2, instead of just copying most of the text from Kulesza & Taskar (2010).
4. Which part of the “DPP kernel built” is learned? The similarity matrix is defined by soft-DTW between prosody feature sequences, and the quality measure is defined by a density estimator (which I assume is the flow module but again was never stated formally).
5. In Sec 4.3, the authors put $logP_L(Y) = det(L) / det(L+I)$, which is inconsistent with Eq.2: it is probability instead of log probability, and the numerator should be $L_Y$ instead of $L$


** Post Rebuttal **
I thank the authors for providing detailed responses to my concerns. After reading the rebuttal and the updated manuscripts carefully, the concerns about clarity and lack of ablation studies have been addressed. However, I still do not agree with the authors' view that neighboring words prosody should be diversified and S-DTW is a suitable similarity metric for the purpose. For utterances with more lively speaking styles, there is a bigger variance of speaking rate and pitch within an utterance, but such variation is typically smooth temporally, instead of bouncing up and down every word, which would resulted in unnatural prosody, which might have been hinted by the worse MOS score in Table 5 when including PDM. In addition, using DTW to align two different spans within an utterance also makes little sense - it should be used to align sequences with the same content but having non-uniform rate distortion. Given these considerations, I would increase the rating from 3 to 5, but I am still in favor of rejection.


**Summary Of The Paper:**

The goal of this paper is to build a text-to-speech model that can generate diverse prosodic features (in particular pitch and duration patterns) without sacrificing the audio quality. The authors propose to cast the prosody selection problem as a determinantal point process, which explicitly takes quality and diversity between elements into account when sampling a subset. To formulate the problem as such, the authors present a prosody diversifying module to generate candidate prosodies for selection, and use soft-DTW as the similarity metric.

**Summary Of The Review:**

The current writeup lacks clarity, ablation studies, and proper evaluations. The way of how conditional DDP is applied also seems problematic.

---

> ### Author Response · Authors · 2021-11-17
> **Response to Reviewer Dfaw**
>
> We thank the reviewer for the detailed comment. Here are our responses.
>
> **Questions/comments**
> >The concept of ground set is never explicitly defined.
>
> - Thanks for pointing this out. How the ground set is constructed is added in Section 4.3 of the revised paper.
>
> > Does the prosody predictor the left and right context separately?
> -  The short answer is NO. The prosody predictor conditioned on both the target and contexts generates prosodic candidates. In addition, the quality of each prosodic candidate is evaluated combined with its neighboring contexts.
>
> > Why conditional DPP is required is not clear when reading Section 2.2.
> - The motivation of using conditional DPP is added in Section 2.2 in the revised
> paper.
>
> > Which part of the ”DPP kernel built” is learned?
> -  We expressed that ”DPP kernel is learned” in the sense that parameters of PDM are learned during the training. The similarity matrix is learned by updating the parameters of PDM.
>
> > The equation in Section 4.3 is inconsistent with Eq.2
> - Thank you for pointing this out. The equation regarding MLE estimation in Section 4 is fixed in the revised paper.
>
> **Weaknesses**
> > Writing requires substantial improvement.
> - Notations are unified in the revised paper. More detailed explanations about training PDM are added in Section 4. Specifically, the process for generating prosodic candidates and construction of the DPP kernel is described in more detail.
>
> > The capability of the model when generating multiple samples is not demonstrated.
> - The capability of modeling diverse samples conditioned on the same text compared to VITS is demonstrated in Section 5.3. However, we didn't generate multiple samples from DPP at once using a greedy algorithm due to the following reasons:  1. The memory and time cost are heavy since $k$ iterations are performed on each target in input text for generating $k$ samples.  2. Unlike other domains, text-to-speech applications require the model to generate just a single speech in practice.
>
> > There is a more serious concern about how conditional DPP is deployed for the subset
> selection
> - If we see each target and context as a phoneme, the target should be coherent
> with the context in terms of duration or pitch. However, if we set each target and context as words, it is desirable that the target and its neighboring contexts have different prosodic
> features for the diversity in a sentence.
>
> > The proposed DPP-TTS also requires tuning the weight of the quality term $w$.
> - It is true that our model also requires tuning the hyperparameter. However, tuning
> the hyperparameter is easier than previous models since the training involves just PDM
> not the whole TTS and the number of parameters of PDM (0.3M) is very smaller than the previous TTS models.
>
> > The MOS result doesn’t support the claim that samples maintain naturalness.
> - MOS evaluations are conducted with newly generated samples. The main difference
> compared to the previous method is that only half of the targets in input text are selected to avoid overlapping between targets and contexts.
> | Model        | MOS-S | MOS-P |
> |--------------|-------|-------|
> | VITS         | 4.08 $\pm$ 0.53 | 3.06 $\pm$ 0.52  |
> | Flowtron     | 3.81 $\pm$ 0.72 | -     |
> | DPP-TTS      | 3.92  $\pm$ 0.58| 3.13 $\pm$ 0.54  |
> | Ground-truth | 4.46 $\pm$ 0.51 |   -    |
>
> > Ablation studies are missing.
> - We add ablation studies comparing DPP-TTS and DPP-TTS without PDM in Section 5.4 of the revised paper. The ablation studies include the MOS test and side-by-side evaluation.

---

> ### Author Response · Authors · 2021-11-30
> **Thank you for your response**
>
> We thank for the reviewer carefully reading our responses and the updated manuscript. Here are our responses regarding the post rebuttal.
>
> **About diversifying prosodic features of neighboring words**
>
> For addressing the monotonous pattern of speech, we have carefully considered the scale or scope of targets of PDM. We think that greater scale (4 ~5 words or a sentence) than noun phrases (usually 2 ~3 words) couldn't resolve the issue because the monotonous pattern can still occur in each target. We believe that diversifying prosody of some neighboring words instead of all neighboring words can achieve the goal while avoiding too much variation in the speech. In addition, we expect that the quality term can address the unnatural up and down of prosody between noun phrases.
>
> **About using DTW to align different time spans**
>
> Yes, DTW have been used for aligning two sequences from the same content. However, we believe that DTW is yet an effective discrepancy for comparing phoneme-level prosodic features from different words. In other words, if two different words (noun phrases in this work) have similar phoneme-level dynamic, DTW takes smaller value than when their dynamics are distinct. We believe that the metric with this property is enough to be used as the similarity metric in this work.

---

### Official Review · Reviewer_djSa · 2021-11-02

**Correctness:** 3
**Technical Novelty And Significance:** 4
**Empirical Novelty And Significance:** 2
**Recommendation:** 5
**Confidence:** 3

**Main Review:**

Strengths:

The paper targets a very relevant problem in speech synthesis. The proposed method involving the PDM and DPP kernel is a novel application of existing methods to the task of speech synthesis. The selected baselines are strong baselines both of which claim to have state-of-the-art performance in speech naturalness while also being able to have diversity in prosody of the synthesized speech. The audio examples demonstrate that the model is able to synthesize diverse speech even though LJSpeech is notoriously neutral in terms of prosody.

Weaknesses:

- The paper seems to oversell the method claiming that the speech naturalness does not suffer (in the abstract and conclusion). The MOS scores and the speech samples do not back up those claims.
- Section 4 is a little hard to understand for someone unfamiliar with DPP literature. The figures really come in handy to understand what is going on in the model. There also seem to be some inconsistency in the text and the figure. Specifically, in "Overview of the PDM process", it is mentioned that PDM takes context and target hidden sequences, however, figure 2 shows that PDM only takes in the target hidden sequence, which makes sense to me. Similarly it is mentioned that the trained prosody predictor takes in the context hidden sequences, but figure 1b shows that the prosody predictor takes in context and target hidden sequences.
- The authors need to better motivate the reason for choosing only noun phrases as the target for prosody diversification.
- Evaluation section has some issues as well:
  - In the side-by-side evaluation the authors find that listeners rate VITS higher than DPP-TTS2 model in terms of prosody variation. Note that this model should actually be more diverse than DPP-TTS1 since we add an additional degree of freedom: pitch. The authors attribute this poor performance to the drop in naturalness, but this is not a valid explanation, since the listers are instructed to rate the prosody diversity and not naturalness.
  - The determinant-based metric is not clear to me. Based on my understanding, first, a square cosine similarity matrix is computed between feature sequences from N generated samples for the same utterance. Then the determinant is computed. Please correct me if I'm wrong. Also, if we look at table 2: there is some inconsistency regarding the determinant metrics. For pitch determinant, the larger value is bold, while for duration determinant, the smaller value is bold. One of these is probably incorrect, unless I have completely misunderstood this metric. Based on my understanding, the larger value should indicate better diversity. Another question though is that these determinant values are very tiny. Are the differences significant, and noticeable? The samples should ideally have also contained a set of samples generated for the same speech which better demonstrates the ability of the model to generate diverse prosody.

Typos, etc:
- Accents in references are not properly rendered. Please check your bib file. E.g. Valles-Peres et al., 2021
- P5: The stochastic and pitch -> The stochastic duration and pitch
- P6: importance sampling equation is missing a closing brace.


**Summary Of The Paper:**

The paper addresses the challenge of synthesizing diverse prosody in text-to-speech systems. Most recent works have now successfully modeled human speech but the delivery usually ends up being monotonous or the average of the training set, and that is the problem that this paper attempts to solve.

The authors propose a prosody diversifying module (PDM) which is based on conditional determinantal point processes (DPP). This module is utilized in tandem with a Fastspeech 2-based TTS model. They modified the base TTS model to incorporate a stochastic duration and pitch predictor. Once this TTS model is trained, the PDM model is trained separately using the prosody feature predictors and text encoder. The authors focus only on noun phrases for where the PDM operates on to diversify prosody. Additionally, the authors use soft-DTW cost to measure the distance between predicted prosody features for context and focused-on phrases which is vital to the construction of the DPP kernel.

The authors compare their approach to other methods of TTS which claim to have prosodic diversity, i.e. VITS and Flowtron. They compare the diversity as well as the naturalness of the samples. The DPP model generally outperforms the baselines in terms of diversity but suffers from loss of naturalness.

**Summary Of The Review:**

The work is well-motivated and novel. The method also achieves diverse prosody which is clear in the samples provided and the MOS scores. However, some of the claims are a little exaggerated. The naturalness does indeed suffer quite a bit. Additionally, there are some lingering confusions that I raise in the review. Thus I am giving a score of 5.

---

> ### Author Response · Authors · 2021-11-17
> **Reponse to Reviewer djSa**
>
> We thank the reviewer for the extensive and detailed feedback. Here are our responses.
>
> **Typo, etcs**
> - Thank you for pointing this out. Typos and misprinted letters are fixed in the revised paper.
>
> **Weaknesses**
> >  Paper seems to oversell the method claiming that the speech naturalness does not suffer.
> - MOS evaluations are conducted with newly generated samples. The main difference
> compared to the previous method is that only half of the targets in input text are selected to avoid overlapping between targets and contexts.
> | Model        | MOS-S | MOS-P |
> |--------------|-------|-------|
> | VITS         | 4.08 $\pm$ 0.53 | 3.06 $\pm$ 0.52  |
> | Flowtron     | 3.81 $\pm$ 0.72 | -     |
> | DPP-TTS      | 3.92  $\pm$ 0.58| 3.13 $\pm$ 0.54  |
> | Ground-truth | 4.46 $\pm$ 0.51 |   -    |
>
> > Section 4 is a little hard to understand.
> - Figures describing the training of PDM are updated in the revised paper. The
> process of constructing the DPP kernel and training of PDM is described in more detail in
> the revised paper. PDM and the prosody predictor are conditioned on both the target and contexts.
>
> > The authors need to better motivate the reason for choosing only noun phrases as the
> the target for prosody diversification.
> - The reason for choosing phrases instead of phonemes or sentences is added in Section 4.1 of the revised paper. We found that noun phrases are less varying in their lengths
> compared to other phrases. Since too much variation of lengths of each target makes the
> training more challenging, we’ve chosen the noun phrase as the target.
>
> > Using both duration and pitch predictors shows inferior results.
> - Side-by-Side evaluations are also newly conducted and generated paragraphs
> outperform VITS model. Regarding the previous results, we believe that it is hard to completely disentangle the naturalness perspective from the evaluation although listeners are
> instructed to consider only the prosody diversity
>
> > The determinant-based metric is not clear.
> - You are right about the determinant-based metric, apologies for the confusion. Metrics are computed with newly generated samples in the revised paper. Since the
> determinant equals the product of eigenvalues of the cosine-similarity matrix, the metric tends to get smaller as the matrix size increases. The samples regarding Section 5.3 are updated
> in the supplementary material.

---

### Official Review · Reviewer_kefC · 2021-11-03

**Correctness:** 3
**Technical Novelty And Significance:** 3
**Empirical Novelty And Significance:** 2
**Recommendation:** 3
**Confidence:** 4

**Main Review:**

The paper tackles an important and practical problem of generating expressive speech. Using DPPs to improve the diversity of predicted prosody is interesting; however, the subjective measures don't demonstrate the effectiveness of the proposed approach. In Table 1, the proposed method comes last in naturalness against the other two baseline systems. Although figure 4 shows that the proposed system can generate more diverse prosody, it might not be naturally diverse. Using both duration and pitch predictors shows inferior results on the right side of figure 4-b. More work is needed to figure out a good recipe for applying DPP to the problem at hand and ensure all components contribute constructively to the overall goal of generating natural and diverse speech.

Another critical area for improvement is writing. The TTS problem and its components are disconnected in presentation from the technical DPP components. Presenting the baseline TTS system and its prosodic components before DPP allows a more integrated presentation of the DPP section motivated from the problem of interest rather than done abstractly. Figure 3 should be brought earlier to clarify what you mean by context and how you are using the Spacy library for noun phrase segmentation. The authors need to unify the symbols used in the DPP sections and TTS sections to enable the reader to understand the proposed approach. The figure font sizes need to get larger, and the figures themselves need to be redone to be more printing friendly.

**Summary Of The Paper:**

This paper addresses the problem of generating expressive speech using Determinantal Point Processes (DPP). Compared to a baseline Fastspeech2 baseline, the authors extend the duration and pitch predictors with a DPP model to favor more diverse candidates. Given that such candidates are variable-length sequences, comparing them is conducted through the differentiable soft-DTW algorithm. The paper shows Mean Opinion Scores (MOS) comparing the proposed approach against the ground-truth audio and two strong models for naturalness. Subjective studies were also conducted to measure the system's ability to generate more speech variability than baselines.


**Summary Of The Review:**

This paper presents an exciting approach to improving the prosodic diversity of generated speech; however, more work is needed before bringing this research to the publication stage. The paper needs a complete rewrite to integrate the technical approach and the problem of interest and use unified symbols across different sections.

---

> ### Author Response · Authors · 2021-11-17
> **Response to Reviewer kefc**
>
> We thank the reviewer for the great feedback. Here are our responses.
>
> >The subjective measures don’t demonstrate the effectiveness of the proposed approach.
> - MOS evaluations are conducted with newly generated samples. The main difference
> compared to the previous method is that only half of the targets in input text are selected to avoid overlapping between targets and contexts.
> | Model        | MOS-S | MOS-P |
> |--------------|-------|-------|
> | VITS         | 4.08 $\pm$ 0.53 | 3.06 $\pm$ 0.52  |
> | Flowtron     | 3.81 $\pm$ 0.72 | -     |
> | DPP-TTS      | 3.92  $\pm$ 0.58| 3.13 $\pm$ 0.54  |
> | Ground-truth | 4.46 $\pm$ 0.51 |   -    |
>
> > Using both duration and pitch predictors shows inferior results.
> -  Side-by-Side evaluations are also newly conducted and generated paragraphs
> outperform the VITS model. Note that **same** samples as the MOS test are used to demonstrate that the samples capture both the naturalness and diversity.
>
> > The writing needs improvement. The method used in this paper is ambiguous.
> - In the revised paper, an explanation about the role of PDM is added in Section 3 to give a more integrated presentation of the DPP section. Symbols used in the DPP sections and TTS sections are unified. Figures are completely redone.

---

### Author Response · Authors · 2021-11-18
**The revised paper is uploaded.**

We have uploaded a revised paper to reflect the reviewer's comments.
Thank all the reviewers for their extensive and constructive comments to make our paper complete.

The updated version includes:
-  Section3 has been reorganized to incorporate PDM into the TTS system and explain the role of each module in DPP-TTS.
-  Section4 has been revised to give a more detailed description of training PDM.
-  We added reconducted MOS and side-by-side evaluations with newly generated samples included in the supplementary material.
-  We added an ablation study compared with DPP-TTS without PDM in Section 5.4.
-  We added audio samples for Section 5.3 in the supplementary material.
-  Section 2.3 regrading Soft-DTW has been slightly modified and the notation for log-likelihood in Section 4.3 has been changed as $\pi(x)$.
 (Last edited Nov 22 20:39 AOE)

---

### Comment · Area_Chair_922j · 2021-11-20
**Please read responses from the authors**

Dear reviewers,

Please read the detailed responses from the authors. How do they change your evaluation? Do you still have major concerns? Thank you.

---

### Decision · Program_Chairs · 2022-01-20

**Decision:**

Reject

**Comment:**

This paper proposes the use of the determinantal point process to introduce the diversity in the prosodic features, including intonation, stress, and rhythm, in text to speech synthesis.  The proposed approach is certainly new, but the experimental support is of critical importance for this work.  One of the major points of discussion was the reliability of the experimental results.  In the original submission, the mean opinion score (MOS) of the proposed approach was inferior to the baseline.  The authors updated the experiments, which significantly (more than the confidence interval) lowers the MOS of a baseline.  This however makes the experimental results questionable.